# Translatomics: The Global View of Translation

**DOI:** 10.3390/ijms20010212

**Published:** 2019-01-08

**Authors:** Jing Zhao, Bo Qin, Rainer Nikolay, Christian M. T. Spahn, Gong Zhang

**Affiliations:** 1Key Laboratory of Functional Protein Research of Guangdong Higher Education Institutes, Institute of Life and Health Engineering, College of Life Science and Technology, Jinan University, Guangzhou 510632, China; qzhaojing@163.com; 2Institut für Medizinische Physik und Biophysik, Charité-Universitätsmedizin Berlin, Charitéplatz 1, 10117 Berlin, Germany; qinbo@qibebt.ac.cn (B.Q.); rainer.nikolay@charite.de (R.N.); christian.spahn@charite.de (C.M.T.S.)

**Keywords:** translatomics, RNC-mRNA, translation regulation, RNC-seq, Ribo-seq, polysome profiling

## Abstract

In all kingdoms of life, proteins are synthesized by ribosomes in a process referred to as translation. The amplitude of translational regulation exceeds the sum of transcription, mRNA degradation and protein degradation. Therefore, it is essential to investigate translation in a global scale. Like the other “omics”-methods, translatomics investigates the totality of the components in the translation process, including but not limited to translating mRNAs, ribosomes, tRNAs, regulatory RNAs and nascent polypeptide chains. Technical advances in recent years have brought breakthroughs in the investigation of these components at global scale, both for their composition and dynamics. These methods have been applied in a rapidly increasing number of studies to reveal multifaceted aspects of translation control. The process of translation is not restricted to the conversion of mRNA coding sequences into polypeptide chains, it also controls the composition of the proteome in a delicate and responsive way. Therefore, translatomics has extended its unique and innovative power to many fields including proteomics, cancer research, bacterial stress response, biological rhythmicity and plant biology. Rational design in translation can enhance recombinant protein production for thousands of times. This brief review summarizes the main state-of-the-art methods of translatomics, highlights recent discoveries made in this field and introduces applications of translatomics on basic biological and biomedical research.

## 1. Introduction

Proteins execute all kinds of biological functions in life; thus, they are under delicate control. According to the central dogma, of molecular biology, which describes the flow of genetic information from DNA, via mRNA to proteins, the generation of the entire proteome consists of four major regulatory steps: RNA synthesis (including epigenetic and transcriptional regulation), RNA degradation, protein synthesis (i.e., translational regulation) and protein degradation. With the advance of current omics technology, mathematical models and omics measurements demonstrated that the translation regulation accounts for more than half of all regulatory amplitudes, beyond the sum of all other regulations [1]. Therefore, translational regulation is the most important regulatory step in organisms. A number of studies focusing on the translation process were carried out since the 1970s but the investigation on global scale was achieved only in recent years. Similar to “genomics” for the whole genome, “transcriptomics” for all transcripts, “proteomics” for all proteins, the nomenclature “translatomics” was raised for the studies on all elements involved in translation.

Due to technological difficulties, translatomics received little attention for some time. First, both nucleic acids and proteins are involved in translational regulation, increasing the complexity and diversity of biological macromolecules involved in the translation process. Various omics tools and skills are needed for translatomics studies. Second, the translation machinery is highly complex and the response to environmental and physiological changes requires rapid and specific adaptation of the translation machinery within minutes, rendering the experiments challenging. The lack of studies on translatomics indicates substantial gaps in the understanding of the most important regulatory step in the flow of genetic information. In recent years, continuous development has been made in the field of translational regulation, allowing us to study the features of translation in comprehensive approaches.

In this review, we will give a brief introduction to the state-of-the-art methods of translatomics to interrogate the multifaceted translational control and the significance and application of translatomics.

## 2. Methods for Translatome Research

The generalized definition of translatome includes all elements that are directly involved in the translational process, such as mRNAs to be translated (also known as RNC-mRNAs), ribosomes, tRNAs, some regulatory RNAs (miRNA, lncRNA and so forth. Note that not all regulatory RNAs are involved in translation), nascent polypeptide chains and various translation factors. Usually the term “translatome” represents the entirety of translating mRNA. In general, RNA and protein are two major categories of macromolecules in translatomics studies. Their static components and dynamics are regulated by a complex and sophisticated system. For each element involved in translation, specific methods have been developed to investigate it in global scale.

### 2.1. Method for Translating mRNA

The mRNA provides the blueprint for protein synthesis. Investigating the translating mRNA is the primary task of translatomics. Since the ribosomes non-covalently bind with mRNA, the ribosome nascent-chain complex (RNC) is very fragile and prone to dissociation or degradation after cell lysis. Several important and classic methods were developed to analyse different features of translating mRNAs: polysome profiling, full-length translating mRNA profiling (RNC-seq), translating ribosome affinity purification (TRAP-seq) and ribosome profiling (Ribo-seq). The basic principle of these methods are illustrated in Figure 1.

#### 2.1.1. Polysome Profiling

Polysome profiling was developed in the 1960s, based on sucrose gradient ultracentrifugation. Ribosome is the largest macromolecular machinery in most cells with high density. An mRNA molecule bound by more ribosomes would sediment faster in sucrose gradient. Therefore, after sucrose density gradient centrifugation, free RNA and proteins float on the top of the sucrose gradient due to their density. The sucrose solution was pumped slowly from the bottom and the mRNAs bound to different number of ribosomes can be separated. The mRNAs in the fractions are then analysed using northern blot, microarray or RT-PCR to reflect the distribution of translation of transcripts. The technique was commonly used in detecting large changes in translation. For example, a significant increase in the composition of a single ribosome was observed under hyperosmotic pressure [2,3], the number of ribosomes bound to a single mRNA increases significantly under oxidative stress [3].

To be noted, researchers thought that actively translated mRNA usually binds multiple ribosomes. However, recent studies have shown that translation are active for those mRNAs that bind single ribosomes. For example, in HEK293 cells and exponentially growing *Escherichia coli* (*E. coli*), which are recognized to be very active in translation, the monosome fractions (70S for prokaryotes and 80S for eukaryotes) are dominant [3,4]. When the monosome fraction was separated and transferred to a cell-free translation system, the ribosomes can resume translation and produce proteins [5]. In *Saccharomyces cerevisiae*, the monosomes are elongating, not initiating. Short open reading frames (ORFs), fast-translating genes and the low-abundance mRNAs tend to be enriched in the monosome fraction [6]. These results proved the translational activity is not proportional to the number of ribosomes bounded on the mRNA.

The main drawback of the polysome profiling is the difficulty in performing in-depth analysis of the all translating mRNAs (RNC-mRNA). Due to the large volume of the sucrose gradient, the RNC-mRNA concentration in each fraction is very low and high concentration of sucrose inhibits some further enzymatic reactions [7]. The total amount of RNC-mRNA recovered from sucrose gradient is generally only enough for RT-PCR quantification but difficult to acquire enough amount mRNA for full-spectrum analysis such as microarray or RNA sequencing unless using large amount of starting materials [7,8].

#### 2.1.2. RNC-Seq

The full-length translating mRNA sequencing (RNC-seq) [9], exhibits unique advantages and effectively resolves this problem. The cell lysate is loaded onto a 30% sucrose cushion and ultracentrifugation is performed to separate all the translating mRNA associated with ribosomes from free mRNA and other cellular components. The RNCs are sedimented by ultracentrifugation. The RNC-mRNA can be easily recovered from the pelleted RNC and effectively avoid the interference of high concentration of sucrose facilitating downstream studies. Next-generation sequencing of RNC-mRNA by this technique reveals the full-length information of translating mRNAs, including the abundance and types. By optimizing the centrifugation and the sucrose cushion, the RNC recovery rate can reach 90% and the recovered RNC retains translation activity when proper buffer was given [5]. The technical difficulty of RNC-seq lays in the separation of intact RNC. The fragility of RNC leads to ribosome dissociation and mRNA breakage/degradation, which result in biased analyses of RNC-mRNAs.

#### 2.1.3. Ribo-Seq

Ribosomal profiling (Ribo-seq), first published by Ingolia et al. in 2009 [10,11], investigates translation from another point of view. Cell lysates were treated with a low concentration of ribonuclease (RNase) to degrade mRNA with exception of those RNA fragments protected by ribosomes. Then, the 22–35 nt mRNA fragments (known as ribosome footprints, RFPs, which are equivalent to ribosome protected fragments, RPFs) were analysed by next-generation sequencing (NGS) to reveal ribosome positions and densities. Based on the positional information, the distributions and densities of ribosomes on each transcript it is possible to deduce information such as the start codon position (including non-ATG initiation), codon usage bias [12], upstream ORFs (uORFs) [13] and translational pausing landscape [14] (see reviews [15,16]). These aspects could not be investigated by other translatomic methods. In 2016, super-resolution ribosome profiling, an optimized method of Ribo-seq was developed, which enabled to observe the robust global 3-nt periodicity in individual transcripts. In addition this method improved the ability of Ribo-seq to uncover small ORFs (sORFs) and unannotated new coding regions likely coding proteins from annotated noncoding RNAs and pseudogenes [17]. Ribonuclease used in ribosome profiling was also take into consideration and found ribonuclease T1 was the best enzyme that can preserved the integrity of ribosome while converting polysomes to monosomes in examined species [18]. A variant of Ribo-seq is the translation complex profile sequencing (TCP-seq) [19,20]. Formaldehyde was used to crosslink the snap-chilled yeast cells, by which any translation complex type to mRNA was stalled at their native positions, followed by RNase digestion. Full ribosome and small subunit (SSU) were then separated using sucrose gradient ultracentrifugation and the RNA fragments up to 250 nt attached to these fractions were then sequenced to generate native distribution profiles during the initiation, the elongation and termination stages of translation, respectively. This method enables observing the SSU footprints along 5′ untranslated regions (UTRs) of mRNA and captures the positions of any type of ribosome-mRNA complex at all phases of translation.

However, Ribo-seq is complicated and costly. It requires large number of cells as starting material, compared to the RNC-seq. Since the hybridization-based rRNA depletion is necessary for Ribo-seq of all species, Ribo-seq is usually restricted to several model organisms. For many other species, especially the large variety of bacteria, Ribo-seq is difficult to perform due to the lack of rRNA probes for rRNA depletion, where the costly customization of rRNA probes is seemingly the only choice. Many other factors affect the number of RFPs such as pseudoRPFs. Since Ribo-seq mainly analyse coding sequence (CDS), where ribosomes bind mRNA. Untranslated regions (UTRs), which can be highly correlated with translational regulation, cannot be efficiently analysed [7]. In addition, Ribo-seq often generates many “RFPs” which are mapped to non-coding RNAs, indicating that a considerable false positive rate [21].

Another drawback is the short length of the RFPs (24~26 nt in prokaryotes and 28~30 nt in eukaryotes), which is restricted by the size of ribosomes and cannot be further extended. In order to obtain sufficient coverage of medium-abundance mRNAs, exaggerated sequencing throughput is required (usually >100 million reads per sample), which means high sequencing and computational cost. Even so, many translation events, especially the splice junctions of splice variants and circular RNAs are still hard to cover. The spliced mapping algorithms are less effective on these short reads to detect junctions. In contrast, the full-length RNC-seq sequences the entire mRNA; therefore, longer read length is applicable. Longer reads result in almost full coverage of most translating mRNAs, including the low-abundance mRNAs [22,23]. This allows efficient detection and quantification of the junctions, for example, the various splice variants of the translating BDP1 and BRF1 [9] and the translating circular RNA circLINC-PINT [24], which were almost unrealistic using Ribo-seq.

It should be emphasized that the density of RFPs is NOT representing translational activity. The RFP density is proportional to the translation initiation rate and inversely proportional to elongation velocity [25]. If the translation is completely stalled on a certain mRNA, the RFP will be highly enriched in this mRNA but the translational activity is zero.

#### 2.1.4. TRAP-Seq

Ribosome affinity purification (RAP) or translating RAP (TRAP) was reported by Inada, et al. [26], using the large ribosomal subunit protein Rpl25p produced under the control of a tissue-specific promoter and a fused affinity tag (such as polyhistidine, green fluorescence protein (GFP), etc.) at the C-terminus [26]. These ribosomes are then purified with affinity (beads or columns) to be separated from the ribosomes of the other cell types. TRAP-seq specifically enriches RNC-mRNA in a sample that is difficult to isolate for subsequent analysis [27] and the ribosomes isolated by TRAP-seq should not be contaminated with non-ribosomal mRNPs that might co-sediment with ribosomes because TRAP-seq does not utilize ultracentrifugation. TRAP-seq has its special advantages in complex tissues to isolate translating mRNAs from specific cell type [28]. However, TRAP-seq needs a stably transfected cell line to produce the tagged ribosomal protein. When applied to plants and animals, constructing stably transgenic organisms is inevitable. This is time-consuming, costly and is not applicable to species, for which no stable transformation method has been established [27,29]. Moreover, overproduction of tagged ribosomal proteins has the potential to alter the structure and properties of these ribosomes. Consequently, the system is no longer under physiological conditions; all conclusions should be carefully assessed before applied to general scenarios.

The technical comparison of these methods are listed in Table 1.

### 2.2. Methods for tRNAome

As essential components of translation process, tRNAs recognize codons on mRNA and transporting corresponding amino acids for protein synthesis. The types and amounts of tRNAs highly influence the speed of protein synthesis [3,25,30]. Numerous tRNA genes have been predicted in genomes (89 tRNA genes in *E. coli* but 595 in hg19 *human* genome, 800 in *Oryza sativa* and up to 13105 in *zebrafish*, according to GtRNAdb [31]).

Various kinds of tRNA molecules are highly homologous in nucleotide sequences: the primary sequence deviation between two different tRNAs can be as low as 1 nucleotide. All tRNA species share highly similar and highly thermodynamically stable secondary and tertiary structures and physicochemical properties. Their nucleotides are highly modified compared to other RNA types. These features make separation and quantification of all tRNA species extremely difficult. Dong et al. [32] and Kanaya et al. [33] isolated and quantified tRNAs in *E. coli* and *Bacillus subtilis* (*B. subtilis*) by 2-dimensional electrophoresis (2-DE), respectively. Mass spectrometry (MS) was also used to identify tRNA species but the extreme homology of tRNA sequences leads to insufficient signatures to distinguish each tRNA species, even in prokaryotes. Kanduc et al. separated 269 rat tRNAs into 62 peaks using high performance liquid chromatography (HPLC) [34]. Resolution of mass spectrometry and hybridization-based microarray were severely restricted to 31~70 tRNA types, due to the extremely high homology of the tRNA sequences (reviewed in Reference [35]). NGS seems to be the only high-resolution method to perform the tRNAome identification and quantification. However, the highly stable structures and the heavily modified nucleotides lead to low and unstable efficiency of reverse transcription and high error rate in the sequencing reads. In 2015, Zhong et al. [3] performed the first tRNA-specific NGS for the *E. coli* tRNAome. They used a primer designed for CCA-tail of tRNAs to prime reverse transcription universally for all tRNAs and performed reverse transcription at higher temperature to counteract the stable structures and modifications, enabling efficient reverse transcription. The mismatches in the reads due to the modifications were tolerated by the highly accurate and robust *FANSe* algorithm [36,37]. Shortly after that, AlkB-facilitated RNA methylation sequencing (ARM-seq) and demethylation (DM)-tRNA-seq was developed, using modified demethylase *AlkB* from bacteria to remove the methyl modification in tRNA and then sequenced the full-length tRNA [38,39].

Hydro-tRNAseq, based on partial alkaline hydrolysis of tRNA, was also used to cleavage tRNAs into shorter reads, resulting in fewer errors at modification sites per sequence read, largely improved the tRNA read content in NGS datasets [40]. However, tRNA anticodons and mRNA codons do not show a one-to-one correspondence. It is impossible to predict tRNA usage merely from mRNA codon usage from mRNA sequence obtained from ribosome profiling. Ribo-tRNA-Seq was developed to investigate this unknown area [41].

Various new developments enabled quantification of tRNAome for any species and the ability to decipher the functional roles of tRNAs under translational control.

### 2.3. Methods for the Folding State of Nascent Polypeptides

Traditional mass spectrometry methods are inappropriate to resolve the structure of nascent polypeptide sequence due to limited sensitivity and sequence coverage. Currently, there is no method for detecting the structure of nascent polypeptide chains at global scale but it is possible to interrogate the nascent chain´s conformation of a single protein. A general and convenient method is the limited protease digestion. The intact RNC is treated by a non-specific protease such as proteinase K at low temperature (Figure 2A). During the protease treatment, the flexible parts of the protein are easily digested, while the tightly folded elements are less accessible to the protease and thus remain uncleaved. The cleavage products can be analysed by gel electrophoresis or autoradiography to reveal the folded regions of the nascent chains [5,30].

The analysis of the fine structure of the nascent polypeptide chain in RNC is demanding. Despite technical advance in both X-ray diffraction and cryo-electron microscopy (cryo-EM) methods these techniques fail to provide high-resolution structures of nascent chains directly. Since protein folding may follow multiple pathways in the folding landscape, nuclear magnetic resonance (NMR) spectroscopy appears to be more suitable for analysing the structure of nascent polypeptide chains due to its ability to determine the dynamic structure of biological macromolecules in solution. In 2006, Hsu et al. used NMR to resolve the nascent polypeptide chain conformation of Ig2 protein [42]. In addition, structural information of nascent α-synuclein was successfully obtained in cultured bacteria, using ^1^H and ^15^N for metabolic labelling [43]. The labelling with stable isotopes eliminates the signals of the large ribosome.

### 2.4. Methods to Identify and to Quantify Nascent Peptides

Identification and quantification the nascent chain is also needed in monitoring the protein synthesis. Although this could be partially replaced by RNC-seq and Ribo-seq, direct evidence at protein level is also preferred. Typical techniques include pSILAC, BONCAT/QuaNCAT and PUNCH-P (illustrated in Figure 2B).

The SILAC (Stable Isotope Labelling by Amino acids) uses the stable isotope added into the cell culture medium to label the newly-synthesized proteins [44]. Mixed with the non-labelled proteins, the mass shift in the mass spectra allows direct comparison of the quantity of the two versions of the same peptides, thus provides accurate quantification. Transferring cell culture to the heavy-isotope medium and start a time-course have been applied to track the protein half-life and turnover [3,45]. However, this measures the accumulation of newly-synthesized proteins, not the nascent peptides. pSILAC (pulsed-SILAC) uses pulse incubation of cell cultures with stable isotopes. If the pulse time is short enough, the heavy-isotope-labelled peptides would represent the nascent chains. However, accurate SILAC quantification requires similar abundance of the labelled and non-labelled peptides. Therefore, pSILAC requires prolonged pulses (>10 h) to achieve sufficient labelling efficiency [46]. This is order of magnitude longer than the translation process of a normal protein (minutes scale), resulting in inaccurate quantification of nascent peptides.

BONCAT/QuaNCAT (Bio-Orthogonal/Quantitative Non-Canonical Amino acid Tagging) [47,48] pulse-label newly-synthesized proteins using azidohomoalanine (AHA), a methionine analogue that allows click-chemistry. Tagged proteins are then isolated and subjected to MS analysis. However, this method requires depletion of the cellular methionine and addition of AHA, creating a stress to the cell and may alter the global translation dynamics and kinetics [49]. Some small proteins would not contain methionine (the initiator methionine may be cleaved when the N-terminal of nascent peptide stretches out of the ribosomal tunnel) and thus cannot be labelled by AHA. Moreover, similar to the pSILAC, the AHA pulse also needs prolonged time (hours) and thus cannot represent genuine nascent chains [50]. The identification power of BONCAT is also limited: 7414 BONCAT-labelled proteins were detected [50], which was much less than the 9064 proteins identified using normal MS in comparable cell lines, published in the same year [51].

To isolate genuine nascent peptides that are being synthesized, PUNCH-P (PUromycin-associated Nascent CHain Proteomics) uses puromycin to label the C-termini of nascent peptides and then enriched by conjugated biotin [52]. This method does not disturb the physiology of the cell until the addition of puromycin and therefore provides a snapshot of the global translation scenario at that moment. However, the drawback is still the lower sensitivity and reproducibility compared to the normal MS. A pioneer work of PUNCH-P identified only 3244 proteins with 5% false-discovery rate (FDR), which is inacceptable according to the Human Proteome Project (HPP) Guideline [53]. Therefore, it is not often used for proteome-wide study but useful to study the dynamics of single proteins [54,55]. In addition, the correlation of the PUNCH-P nascent peptides and the Ribo-seq RFP abundances are generally very low (*R* < 0.38) except very high abundance proteins, suggesting high variability and bias of such methods [56].

With the advancement of MS experimental methods and bioinformatics, especially using the RNC-seq as an independent reference against the proteome at steady state, the sensitivity and confidence of MS have been substantially improved [22,23,51,57]. Combining multiple techniques, we have used label-free MS method to identify 4140 nascent peptides in *E. coli*, which is approaching the sensitivity of Ribo-seq results (identified 4195 translating genes by RFPs) [58].

### 2.5. Methods for Detecting mRNA Co-Translational Decay Intermediates

To maintain dynamic gene regulatory networks in cells, mRNA decay is an important process in controlling mRNA abundance at posttranscriptional level. It is generally assumed that normal mRNAs undergoing translation are stabilized and protected from decay by a 5′-m7GpppN cap and a 3′-poly-A tail. Moreover, recent report have revealed that mRNA decay can be coupled with translation [59,60]. In general, mRNA degradation is mediated by three main mechanisms: 5′-to-3′degradation, 3′-to-5′decay and small RNA-mediated RNA silencing, which were different from degradation of aberrant mRNAs by other three pathways—non-stop decay (NSD), nonsense-mediated mRNA decay (NMD) and no-go decay (NGD) [61]. In RNA silencing, the small RNAs including microRNAs (miRNA) and small interfering RNAs (siRNA) can initiate argonaute (AGO)-mediated cleavage of target RNAs in special positions with complementarity base-paring interactions, which result the mRNA with a 5′-monophosphate and then the target mRNA was degraded. Different high-throughput sequencing methods based on ligation of 5′-monophosphate such as genome-wide mapping of uncapped and cleaved transcripts (GMUCT), parallel analysis of RNA ends (PARE), 5′-monophosphorylated ends sequencing (5Pseq) and degradome sequencing and so forth have been developed to specifically study the degradation of mRNAs [62,63,64,65].

GMUCT is useful to study genome-wide mapping of uncapped transcripts and miRNA-directed target RNA cleavage and co-translational RNA decay in eukaryotes transcriptomes [66]. PolyA-containing mRNAs were isolated and an adapter is ligated to the free 5′-monophosphate ends in mRNAs, intermediate and then the NGS library was generated and sequenced. GMUCT was further used to investigate ribosome pausing during termination, uORFs finding during co-translational decay [67]. PARE is also developed to identify the miRNA cleavage sites [63] and to investigate the ribosome stalling and the identification of translation regulators [68]. 5Pseq direct ligate 5′-monophosphorylated(5P) ends of decapped transcripts and mRNAs from the same sample are treated with a phosphatase to block the 5P end of decapped transcripts and the capped mRNAs were captured and sequenced in parallel. By compare the capped and decapped sequencing samples, the location of mRNA degradation intermediates could be revealed. 5Pseq can also reveal ribosome dynamics such as ribosome pausing and termination [69]. Note that, all those methods above can only be used to measure the miRNA-mediated mRNA cleavage, NMD and 5′-to-3′degradation but cannot measure 3′-to-5′ decay of the mRNA intermediate. And miRNA target cleavage sites can be detected in plants more obviously than in animal and human, because the different miRNA silencing mechanisms [66].

### 2.6. Visualization of Translation In Vivo

The abovementioned methods require lysis of the cells. To monitor live process of translation, single molecule fluorescence resonance energy transfer (FRET) was employed to determine the reaction rates within the elongation cycles and found that unfolding of mRNA secondary structures is more closely coupled to E-site tRNA dissociation than to tRNA translocation [70]. FRET was also used to identify ribosomes translating a given mRNA from a mixture of two different mRNAs [71]. Nascent chain tracking (NCT) can monitor and quantify the protein synthesis dynamics at the single mRNA level by multi-epitope tags and antibody-based fluorescent probes [72]. By employing NCT, an elongation rate of ~10 amino acids per second were revealed, with initiation occurring stochastically every ~30 s. However, these methods are not high-throughput compatible.

## 3. Translatomics in Fundamental Biology

### 3.1. The “Quantitative” Central Dogma of Molecular Biology

The quantitative correlation between mRNA and protein abundances in various species is generally poor (*R*^2^ ~ 0.01–0.5) [73]. This means that mRNA levels are not sufficient to predict protein levels in many scenarios [74] because translation control plays a major role in the flow of genetic information from DNA, via mRNA, to polypeptides, according to the *central dogma of molecular biology.* In 2013, Wang et al. discovered multivariate linear correlation among the abundance of translating mRNA, mRNA length of protein abundance in steady-state cells with *R*^2^ = 0.94 by employing mRNA-seq, RNC-seq and mass spectrometry [9]. It quantitatively connected the key steps of the genetic flow from transcripts to functional proteins. Taking the advantage of the high sensitivity and precision of NGS, RNC-seq may partially replace proteome quantification in steady-state cells in some cases.

### 3.2. Translational Pausing Induces Co-Translational Folding

Any protein synthesized by ribosomes must be folded into a specific three-dimensional structure to fulfil its cellular functions. Anfinsen’s Dogma stated the primary sequence of a protein determines its three-dimensional structural conformation in a given environment. According to Anfinsen’s Dogma, it is generally believed that silent mutations do not alter the amino acid sequence and thus do not affect protein folding. However, one amino acid can be encoded by various codons except for methionine and the concentration of corresponding tRNA can vary by more than one order of magnitude. Therefore, the translation elongation velocity is non-uniform along the mRNA and the silent mutations may alter the elongation velocity. Zhang et al. proposed a mathematical model for predicting the translation elongation velocity based on tRNA concentration [5]. The translation velocity is effectively slowed down by several slow-translating codons. These regions are called “translational pausing” or “attenuation sites,” where ribosomes move slower. Pausing sites typically occur 20–70 amino acids downstream of a protein domain [75], coordinating the fast protein biosynthesis and slow co-translational folding. This mechanism simplifies the folding of a large protein into several smaller domains and solves the Levinthal’s Paradox, according to which a folding protein, sampling all possible conformations, would practically never reach its native state [76]. There are more translation attenuation sites in longer proteins, which contain more domains to fold and thus rely more on pausing-mediated domain-wise co-translational folding [9]. This tRNA-mediated protein folding mechanism is independent of molecular chaperone folding mechanism. Acceleration of these pausing sites by silent mutations or tRNA overexpression can perturb the domain-wise co-translational folding, resulting in misfolding and mislocalization of proteins. This “stop-to-fold” theory explains many diseases caused by silent mutations (reviewed in Reference [77]). Protein orthologs in different organisms share similar pausing profiles, although their codon usage and coding sequences substantially vary. Therefore, translational pausing provides an additional selective force for evolution [75].

### 3.3. Two-Dimensional Translational Control Initiation and Elongation

Translation initiation and peptide elongation are the two major stages of protein synthesis control. The translation ratio (TR) value of each gene can be calculated by measuring the mRNA and RNC-mRNA of the same sample in parallel [9]. In eukaryotes, translation initiation is the main rate-limiting step in protein synthesis; thus, TR approximately reflects the translation initiation efficiency [9]. In prokaryotes, most genes are organized in polycistrons, aggravating the determination of translation initiation efficiency of each individual gene.

To detect the translation elongation speed globally is even more demanding Ingolia et al. measured translational elongation speed in mouse embryonic stem cells using Ribo-seq by inhibiting translational initiation and then tracking the length of the RFP-free mRNA regions [78]. However, an artificial lag of 60 s showed significant bias of this approach, making their approach far from the physiological conditions. Substantial error limited this method in measuring average elongation speed of many genes. The technical flaws lead to an ironic conclusion: there’s no difference on the translational elongation speed of different types of genes.

Lian et al. performed mRNA-seq, RNC-seq and Ribo-seq simultaneously for the same cell lines and obtained the elongation velocity index (EVI) for each individual gene under physiological conditions the first time [25]. tRNA concentration was found the only factor that regulates the translation elongation rate. Various codon preference indices, such as codon adaptation index, codon bias index, tRNA adaptation index and so forth, are irrelevant to actual translation elongation speed. mRNA secondary structure is also irrelevant to the global translation elongation speed.

The TR and EVI, representing translation initiation efficiency and elongation speed respectively, formed a two-dimensional translational control (Figure 3): (I) Most genes are similar concerning TR and EVI. (II) Productivity mode: high TR genes are very actively translated and the folding of these proteins was robust so that no translation pause is required. (III) Quality mode: low EVI genes are slow in translational elongation for better folding to ensuring their functionality. (IV) No gene is active in translation initiation and slow elongation to avoid “ribosome traffic jam”.

Most genes are organized in polycistrons in prokaryotes; therefore, it is still difficult to measure the translation initiation efficiency of each individual genes due to two reasons. First, translation of one gene in a polycistron leads to detection of the entire operon in RNC-seq. Second, besides canonical translation initiation process, a novel initiation mode, “70S-scanning initiation,” has been demonstrated to be ubiquitous in bacteria. The “70S-scanning initiation” represents that the 70S ribosomes do not dissociate after termination step but rather scan along with the mRNA until reaching the initiation site of the downstream cistron of the same mRNA. Binding of fMet-tRNA triggers 70S scanning, which occurs in the absence of energy-rich compounds (e.g., ATP, GTP) and seems to be driven by unidimensional diffusion [79]. The role of IF2 (initiation factor 2) is well defined. It can bind directly to the 30S subunit, providing a docking site for fMet-tRNA [80], while it can also enter the 30S subunit as ternary complex fMet-tRNA•IF2•GTP [81]. Both IF2 and IF3 are essential for cell viability. The binding site of IF3 is at the 30S interface [82], which explains its anti-association effect [83,84], as well as its role in dissociation of the terminating ribosome [85]. Both IF3 and IF2 are also responsible for the fidelity of decoding the initiation AUG at the P site of 30S subunits [86]. IF1 is universal [87] and essential for cell viability [88] although it is the smallest factor, with only 72 amino acid residues in *E. coli*. It binds to the decoding centre at the ribosomal A site [89]. Several functions have been described, for example, stimulation of the formation of the 30SIC and subunit association [90]. It is recently reported that the task of IF1 appears to be the prevention of untimely interference by ternary aminoacyl (aa)-tRNA•elongation factor thermo unstable (EF-Tu)•GTP complexes during the 70S scanning initiation [79].

### 3.4. Alternative Translation Start Sites and Readthrough

Like alternative splicing, which expands mRNA diversity, alternative translation start sites and readthrough expands the protein diversity at the translational level. Specific inhibitors of translation initiation, such as harringtonine and lactimidomycin, enrich the ribosomes at translation start sites. These sites can be analysed by Ribo-seq at genome-wide scale [78,91]. Wang et al. measured the translational status of mRNA isoforms with distinct start sites and revealed nearly 1/10 of the expressed genes in mouse fibroblasts exhibited significant isoform-divergent translation [92]. Isoforms with longer 5′UTRs tended to translate less efficiently [93].

Readthrough, that is, continued translation at the stop codon, produces an extended protein with alternative functions. In addition, readthrough is necessary for many viruses to complete their reproductive cycles. The natural rate of readthrough is less than 0.1% [94]. In rare cases, natural stop suppression increases the readthrough rate by several orders of magnitude [95]. Readthrough can be indicated by RFP distributions after the stop codon [96]. Many readthrough candidates were both identified in yeast and in primary human foreskin fibroblasts, indicating that readthrough might be not only a ubiquitous feature of eukaryotic translation but also a novel mechanism to tune gene expression (reviewed in [97]). However, ribosomes do not dissociate from the mRNA immediately after the translation termination (“run-off”), which can be exaggerated under stress conditions [2,98]. Some translation factors also promote programmed stop codon readthrough, for example, eIF3 [99].

### 3.5. Ribosome Diversity

Canonical stereotype of ribosomes is just a macromolecular machinery for translation. People usually consider that all functional ribosomes in one species are the same. However, recent findings suggest ribosomes are not created equal. Diversity of ribosome composition and activity may be more dynamically regulated to impart a new aspect of translational control (reviewed in References [100,101]). In brief, ribosome diversity may be caused by diversity of rRNA sequences, ribosome protein (RP) composition and translation factors. The rRNA sequence can be sequenced using NGS and the protein components can be resolved using MS.

In rare cases, some species like *Plasmodium* parasites carry two types of ribosomal DNA genes in their genome, generating two types of ribosomes during different stages of their life cycles, respectively [102]. These ribosomes are distinct in function [103]. More common way to construct specialized ribosomes is to generate distinct RPs in different cells and tissues. This includes alternative splicing of RPs [104], cell-specific activation of RP homologs [105], development stage-specific modification of RPs [106,107], sex-specific RPs [108] and tissue-specific transcriptional regulation of RP genes [109]. Ribosome-associated factors also affect ribosome function. For example, Reaper protein inhibits cap-dependent translation and promotes IRES-mediated translation [110].

## 4. Translatomics in Biology/Disease-Relevant Studies

### 4.1. Perturbation of Global Translation in Cancer

Cancer is among the leading causes of death worldwide in high and middle-income countries. The genome instability drives various mutations and leads to drug resistance. Therefore, any treatment targeting genomic variation would be eventually invalid.

Wang et al. found that the translation initiation efficiency (TR), calculated from RNC-seq and mRNA-seq results, is generally elevated in lung cancer cells compared to normal cells [9]. The functions of top 123 TR up-regulated genes reflect almost all hallmarks of cancer, indicating that the preferentially translated genes determine the function and phenotype of the cells. Shorter genes (corresponding to smaller proteins) are more actively translated than longer genes in cancer cells. Guo et al. proposed a mathematical model to explain this length-dependent translational priority [111]: synthesis of shorter proteins are more effective due to less errors during the protein synthesis process (which is dominated by folding errors). Therefore, to synthesize a functional proteome rather than a single protein, the length-dependent translation priority towards small proteins is the most energy-efficient way. With increased translation elongation speed in cancer cells, the folding error rate increases, decreasing the effectiveness to produce large proteins. Therefore, elevating TRs of small proteins facilitates the survival of cancer cells.

Lian et al. used mRNA-seq, RNC-seq and Ribo-seq to calculate EVI to evaluate the translation elongation speed at individual gene level under physiological conditions [25]. They found the translation elongation speed of oncogenic genes is significantly reduced in cancer cells to ensure the correct folding and their malignant function [25]. Meanwhile, the translation elongation speed of tumour suppressor genes is accelerated in cancer cells, perturbing their folding and functionality. This bidirectional regulation on translation elongation allows cancer cells to maintain the malignant phenotypes.

These translatomic investigations provide new insights on the molecular basis of cancer, indicating novel therapeutic strategies.

### 4.2. Microbial Stress Resistance

Bacteria need to respond quickly to sudden environmental changes and the accumulation of reactive oxygen species (ROS) is one of the most common stress. To counteract oxidative stress, the oxidative response system in bacteria, such as SoxRS system and OxyR system, is transcriptionally regulated and takes 20–30 min to take effect [3]. An alternative mechanism is needed to cope with stresses within minutes.

Zhong et al. investigated *E. coli* protein synthesis using mass spectrometry and found that the protein synthesis was almost halted immediately after the application of oxidant [3]. Detailed analysis showed that translation initiation rate was not affected but the translation elongation rate was severely slowed down. tRNA-seq, validated by qRT-PCR, exhibited that almost all tRNAs was down-regulated at the beginning of oxidative stress. Overexpression of low-abundance tRNAs accelerated translation, allowing faster growth under oxidative stress and protecting it against higher concentrations of hydrogen peroxide and tolerating ROS caused by ciprofloxacin. Therefore, under normal condition, excessive tRNA levels abolish translation pauses, resulting in protein folding failure; under oxidative stress, higher tRNA levels maintain protein synthesis to replace the damaged counterparts. This study discovered a new oxidation response mechanism mediated by tRNA for the survival and fitness of bacteria, providing a new perspective for bacterial resistance.

*Streptococcus pneumoniae* (*S. pneumonia.*) is one of the important human pathogens. Iron uptake is essential for the survival of S. pneumonia. Three transferrin protein complexes PiaABC, PiuABC and PitABC are already known. Interestingly, knocked out of all 3 systems does not abolish iron uptake, indicating an alternative iron transport system exists. Yang et al. performed RNC-seq to identify up-regulated genes/proteins under iron deficiency stress [112] and found a new iron-binding protein SPD_1609 which allows iron transport. This protein and its function was then experimentally validated. This study established an effective strategy to discover new functional proteins [112].

### 4.3. Rhythmic Translation in Circadian Clock Regulation

Both gene expression in mammalian and plants displays widespread circadian oscillations. Although most core clock factors are transcription factors, post-transcriptional control introduces delays that are critical for circadian oscillations. Transcription rhythmicity overlap poorly to protein rhythmicity at global scale [113]. Therefore, translational control plays a major role in this process. About 150 mRNAs undergoes rhythmic changes in translation efficiency (obtained using Ribo-seq and mRNA-seq) and serve as predictor of footprint rhythms [114]. Castelo-Szekely et al. discovered tissue differences in translation efficiency modulate the timing and amount of protein biosynthesis from rhythmic mRNAs, consistent with organ specificity in clock output gene repertoires and rhythmicity parameters [115].

### 4.4. Translational Control in Plants

Unlike mammals or microbes, plants are immobile. Therefore, they need mechanisms to quickly adapt to a wide variety of environmental stresses, where translational regulation plays an important role. Translation in plants is globally reduced by 50–77% under heat stress [116] or hypoxia [117], while those mRNAs encoding stress response proteins are selectively translated. Bai et al. identified large sets of translational regulated genes during seed germination of Arabidopsis and found changes in polysome occupancy were not uniformly present throughout the germination process but were restricted to two temporal phases, one during encompassing seed hydration and another during seed germination, revealing an additional layer of gene expression regulation and its dynamics during germination [118]. Polysome profiling and ribosome profiling is the most commonly used method. Recently, RNC-seq is being applied to the studies of plant translational control, for example, the important contributors to plant immunity in Arabidopsis [119].

## 5. Application of Translatomics

### 5.1. Missing Protein and New Protein Discovery

The Human Proteome Project (HPP) aims at verifying all the proteins encoded by predicted coding genes [120,121]. Mass spectrometry has become a major technique for proteome research; however, technical limitations of MS and the physical and chemical properties of proteins affect protein detectability, which become a hindrance to determine the whole proteome. Though the draft-map of human proteome have been published in 2014 [120,121], protein products of thousands of genes were still not detected [122,123], including “missing proteins” (encoded by known coding genes but are not detected at protein level) and “new proteins” (or novel proteins, the proteins coded by the “non-coding RNAs”). Translating mRNA provides direct translational evidence, regardless the physical and chemical properties of the proteins and identifies protein variations indirectly. Taking the advantage of the high sensitivity of RNC-seq, translating mRNA showed its power in identifying missing proteins [124,125]. Although with shorter reads and higher bias [126], Ribo-seq also delivered insights with respect to the missing proteins [127,128,129,130]. Nevertheless, Ribo-seq is less sensitive for specific isoforms [131]. Much longer reads of RNC-seq avoid these problems, which makes RNC-seq a powerful tool in missing protein discovery.

In 2014, the Chinese Human Proteome Project completed the multi-omics level analysis of liver cancer cell lines [51]. From each cell line, more than 14,000 translating genes were detected and only 2/3 of the protein products were detected in MS, showing the high sensitivity of RNC-seq [22,51,132,133,134].

Discovering new proteins is one of the unsolved problems in the HPP. RNC-seq has also achieved remarkable results in searching for new proteins. 6-frame translation of the genome would tremendously expand the protein database, increasing the false discovery rate (FDR) and thus decreasing the sensitivity [135]. RNC-seq can provide full-length translation evidence of proteins without relying on annotation databases directly. Therefore, RNC-seq provides an accurate protein database which could exist in the sample without exaggerating the protein database, thus controlling the FDR and maintaining the sensitivity. RNC-seq found 1397 genes that were annotated as non-coding RNA (ncRNA) in RefSeq database [9] and some of them were later confirmed at protein levels [136]. Due to the important contribution to the HPP, RNC-seq has become one of the four core pillars of the HPP [22].

### 5.2. Enhancing Recombinant Protein Production

Recombinant proteins are widely used for pharmaceutical and industrial purposes. Genes of interest are transferred to appropriate vectors, which are delivered to industrial protein expression systems such as bacteria or yeast to allow mass production, for example of insulin. However, most heterologous recombinant proteins and enzymes cannot be expressed in active form because of misfolding [137]. Several traditional methods have been used to overcome this problem, such as renaturation of inclusion bodies, co-expression of molecular chaperones and elevated levels of tRNAs decoding rare codons, which in general turned out to be rather ineffective. According to the translation pausing theory [5], proper translational pausing sites are necessary for most proteins. Due to the different tRNAome in different species, the recombinant gene is not expressed at its optimal speed, leading to protein misfolding [5,75]. An overall decrease in translation rate by lowering the temperature may not be effective, because the folding rate is also temperature-sensitive. Chen et al. first developed the rational design of translational pausing for protein expression [30]. They made synonymous substitutions to Cyanovirin-N (CVN), an antiviral protein originated from cyanobacteria, to create a translational pausing site based on the structure. Without changing the amino acid sequence and the expression conditions, the soluble yield of the pausing-optimized CVN variant was increased more than 2000×-fold compared to the wild type and the specific activity exceeded the natural protein, indicating that the folding is dramatically enhanced by the artificially generated pausing site. Another successful example is the epoxide hydrolase from *Agrobacterium radiobacter*: the rational design of translational pausing increased the solubility by 40% [138]. Molecular dynamics simulation and experimental validation showed that the steady-state structural fluctuation is a predictor of the necessity of pausing-mediated co-translational folding for proteins [139]. These works demonstrated substantial potential of the translational pausing optimization strategy in industrial protein production.

## 6. Internet Resources for Translatome

Translatomics methods are mostly technically demanding and costly. Therefore, public databases and analysis resources are highly welcomed. There are already some static databases of Ribo-seq data online, such as RPFdb [140], GWIPS-viz [141] and RiboSeqDB [142]. However, their datasets are restricted in Ribo-seq. TranslatomeDB collected RNC-seq, Ribo-seq and their corresponding mRNA-seq data to perform uniformed data analysis for TR and EVI [143]. Importantly, TranslatomeDB allows user to upload or specify new data and analyse promptly with the support of powerful cloud-computing clusters. Unified, accurate and robust bioinformatics pipeline allows mutual differential expression analysis for any two datasets of the same species. Besides the abovementioned primary analysis tools and databases, secondary databases for ORFs and translation initiation sites are also available, such as sORFs.org [144] and TISdb [145]. For translational pausing sites prediction, Ribotempo (http://www.translatome.net/RiboTempo/) is an online server for *E. coli* and *B. subtilis* [75], in which the absolute tRNA abundances were resolved. More software is reviewed in Reference [146].

## 7. Conclusion and Perspectives

Translational control is faster and more sensitive than transcriptional control, which greatly improved the flexibility and complexity of gene expression regulation, allowing organisms survive under various stress conditions. Translatomics allow us to directly investigate the most important step at global scale. It brings theoretical breakthrough and leads to new applications in medicine and engineering. Future improvements of these methods would be focused on decreasing the amount of starting materials (e.g., single-cell translatomics), elevating the sensitivity while minimizing false positives, increasing the spatial and temporal resolution, reducing the cost and operational complexity and expanding the methods to more species and tissue types. We believe that translatomics will demonstrate its significance in biology and biotechnology soon.

## Figures and Tables

**Figure 1 ijms-20-00212-f001:**
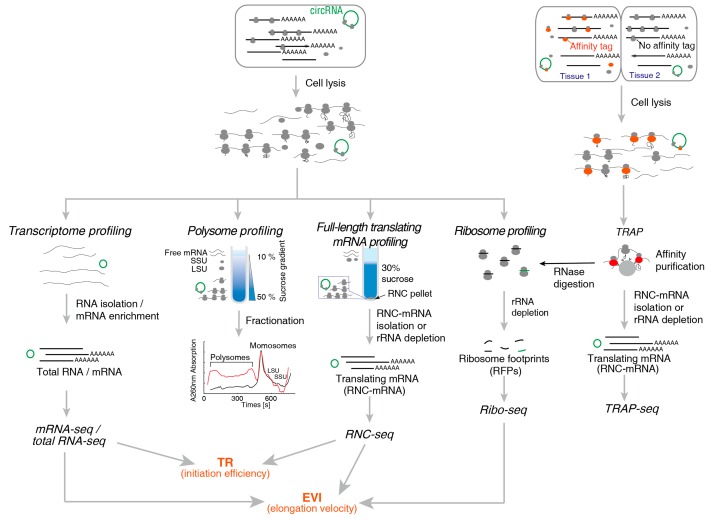
The major translatomic methods which investigate translating RNAs.

**Figure 2 ijms-20-00212-f002:**
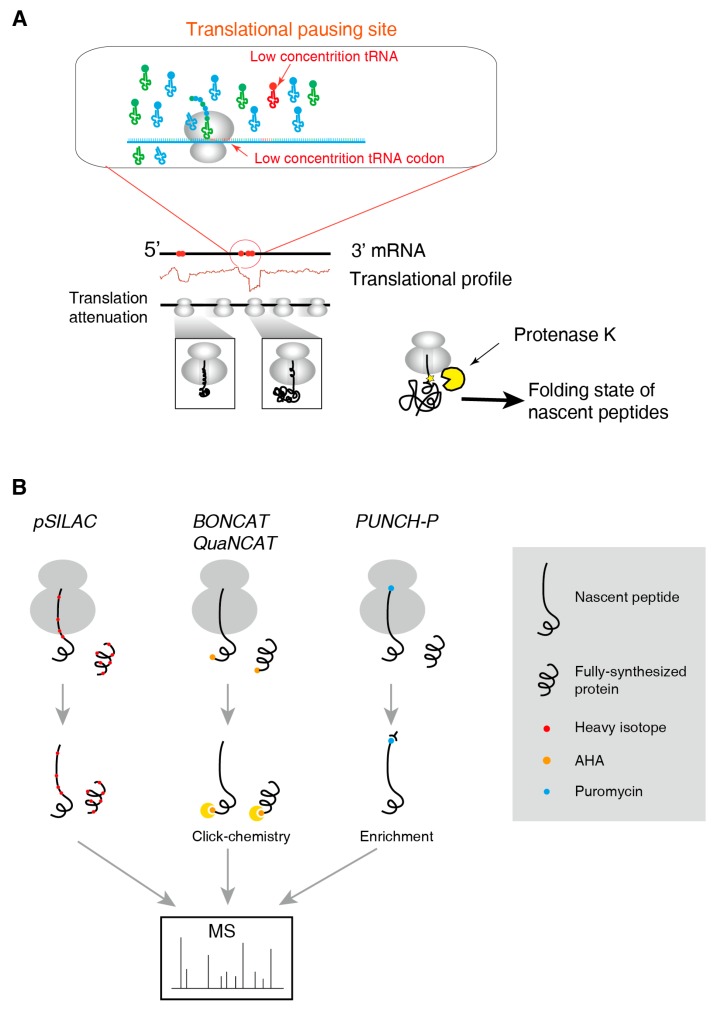
The major methods to investigate nascent peptides. (**A**) Translational pausing sites and the co-translational folding of nascent peptides. Proteinase K limited digestioni is often used to detect the folding state of nascent peptides. (**B**) High-throughput methods to identify and quantify nascent peptides.

**Figure 3 ijms-20-00212-f003:**
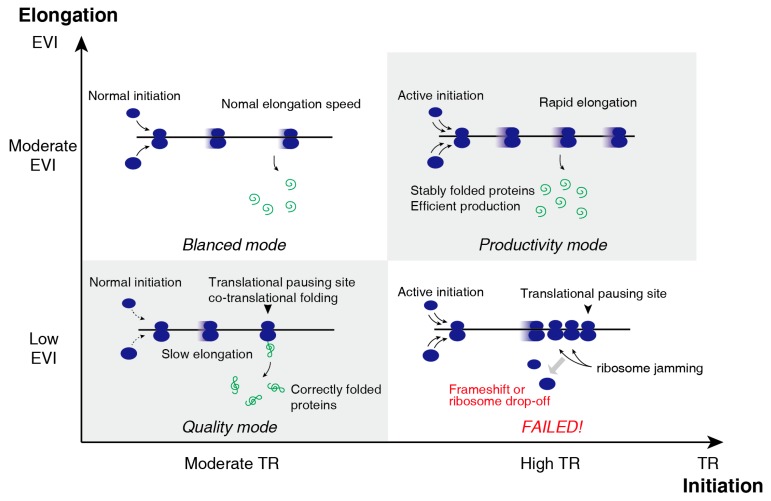
Two-dimensional control mode of translation initiation efficiency (TR) and translation elongation rate (EVI) Most genes are of Balanced mode, with moderate TR and moderate EVI. A small fraction of genes undergoes active translational initiation and rapid elongation and thus are very efficiently produced. These genes under “Productivity mode” do not require co-translational folding for its final conformation; thus, rapid elongation do not impair their functions. Another small fraction of genes is synthesized with moderate TR but low EVI, indicating that there are considerable translational pausing sites along their mRNAs for better co-translational folding quality to ensure their functionality and thus are of “Quality mode.” Simultaneous high TR and low EVI would result in ribosome jamming and thus were eliminated during the evolution.

**Table 1 ijms-20-00212-t001:** Technical comparison of translatomic methods.

Technical Aspects	Polysome Profiling	RNC-Seq	Ribo-Seq	TRAP-Seq
Recovery of RNC-mRNA	Full-length	Full-length	Ribosome protected fragments	Full-length
Difficulty in Recovering translating mRNA	Demanding	Simple	Demanding	Simple
High-throughput methods can be used	Microarray, NGS	Microarray, NGS	NGS	Microarray, NGS
Throughput requirement	Low	Low	High	Low
Read length	Any	Any	22–35 nt	Any
Detecting sequence variations	Simple	Simple	Demanding	Simple
UTR	Yes	Yes	No	Yes
Obtain the position of ribosome, densities, ORF, uORFs	No	No	Yes	No
Obtaining the amount of ribosomes in single mRNA	Yes	No	No	No
Tissue specific	No	No	No	Yes
Under physiological conditions	Yes	Yes	Yes	No
Required experimental steps	Simple	Simple	Complex	Complex

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
