# Peer review of "Translatomics: The Global View of Translation"

_ijms, 2019, doi:10.3390/ijms20010212_

Reviewer 1 Report

The manuscript by Zhao and Zhang is a review on translatomics. The authors describe this field with emphasis on technical advances and recent applications.
In my opinion, the methods are not well described and recent important techniques are missing. This review needs to be improved and more detailed.
Bellow my comments :
For tRNAome :
- Recent techniques like hydro-tRNAseq (Gogakos et al 2017 Cell Reports) or Ribo-tRNAseq (Chen and Tanaka 2018 Cell Reports) are missing.
For translating mRNA :
- It is now well accepted that analysis of decay intermediates could also reveal ribosome dynamics. Recent techniques like 5'PSeq (Pelechano et al. 2015 Cell) or GMUCT (Willmann et al., 2014 Methods) must be described.

- Some improvements of RP exist (e.g. Hsu et al. 2016 PNAS) .

For nascent peptides :
- Some techniques exist for the analysis of nascent peptides (e.g. PUNCH-P). A section on measuring translation via newly synthesized proteins must be provided.

Author Response

Reviewer 1:

The manuscript by Zhao and Zhang is a review on translatomics. The authors describe this field with emphasis on technical advances and recent applications.

In my opinion, the methods are not well described and recent important techniques are missing. This review needs to be improved and more detailed.

Bellow my comments :

For tRNAome :

 - Recent techniques like hydro-tRNAseq (Gogakos et al 2017 Cell Reports) or Ribo-tRNAseq (Chen and Tanaka 2018 Cell Reports) are missing.

Response: We thank the reviewer for this information. We have added description of these methods in section 2.2

For translating mRNA :

 - It is now well accepted that analysis of decay intermediates could also reveal ribosome dynamics. Recent techniques like 5'PSeq (Pelechano et al. 2015 Cell) or GMUCT (Willmann et al., 2014 Methods) must be described.

Response: We thank the reviewer for this information. We have added description of these methods in section 2.5

- Some improvements of RP exist (e.g. Hsu et al. 2016 PNAS) .

Response: We thank the reviewer for this information. We have added description in section 2.1. 

For nascent peptides :

 - Some techniques exist for the analysis of nascent peptides (e.g. PUNCH-P). A section on measuring translation via newly synthesized proteins must be provided.

Response: We thank the reviewer for this point. We have added a section describing the methods analyzing the nascent peptides, including the pSILAC, BONCAT/QuaNCAT and PUNCH-P. We also drew a figure for that.

 Reviewer 2 Report

This short review on an important topic is a positive contribution.

 Author Response

Thank you very much for your appreciation!

Reviewer 3 Report

Authors introduced a summary of method and application of translatomics. The authors introduced several methods for translatome research, tRNAome, and peptide folding detection. In addition, the authors discussed the application of translatomics on fundamental biology and various biological research. In overall, the manuscript properly covered issues in the translatomics study and would be a valuable contribution to the society. However, there are some suggestions that would make this review more appreciable for the research community and the authors should address these issues discussed below.

 Major comments

1) Throughout the manuscript, the authors address RNC-seq to be a better method to recover full-length translating RNA sequences. However, the manuscript does not contain enough comparative results showing the advantageous aspect of RNC-seq. The authors should discuss some results from RNC-seq in more detail.

2) The authors mentioned elements consisting translatome in “Methods for translatome research”. However, including regulatory RNAs in the translatome seems to be inadequate and might mislead to a misunderstanding that all regulatory RNAs are involved in translational process. In addition, the manuscript does not contain any information regarding translation factor. The authors should exclude regulatory RNAs in the given sentence and add information regarding translation factor in translatomics.

3) In table 1, the authors should replace the description Easy/Hard with a more objective information. For example, “Operational difficulty” can be replaced with “Required experimental steps”.

4) The authors should add legend for figure 1 and increase the size of its text.

5) The authors should explain the figure 2 in more detail.

6) In “Application of translatomics”, the authors introduced translatomics application in various fields. It seems that this section could be separated into two. The authors should group 4.1, 4.2, 4.5, 4.6 as tranlsatomics in disease or stress condition for global translational detection and group 4.3, 4.4 as application of translatomics.

7) The authors should describe each study in more detail in terms of the methods used to obtain results mentioned in “Application of translatomics”.

7) In “Conclusion and Perspectives”, the authors should add more discussion regarding future perpective of translatomics and further improvements of the methods discussed in the manuscript.

8) The authors should improve English grammar thoroughly.

Author Response

Reviewer 3:

Authors introduced a summary of method and application of translatomics. The authors introduced several methods for translatome research, tRNAome, and peptide folding detection. In addition, the authors discussed the application of translatomics on fundamental biology and various biological research. In overall, the manuscript properly covered issues in the translatomics study and would be a valuable contribution to the society. However, there are some suggestions that would make this review more appreciable for the research community and the authors should address these issues discussed below.

Major comments

1) Throughout the manuscript, the authors address RNC-seq to be a better method to recover full-length translating RNA sequences. However, the manuscript does not contain enough comparative results showing the advantageous aspect of RNC-seq. The authors should discuss some results from RNC-seq in more detail.

Response: We thank the reviewer for his interest on the RNC-seq versus Ribo-seq. Actually, there are almost no published study which directly compared RNC-seq and Ribo-seq. As a review, it’s not easy for us to present our comparison of RNC-seq and Ribo-seq in this manuscript in detail. Nevertheless, we have written three paragraphs in 2.1.3 (half a page) to discuss the advantageous aspects of RNC-seq against Ribo-seq with a series of citations.

 2) The authors mentioned elements consisting translatome in “Methods for translatome research”. However, including regulatory RNAs in the translatome seems to be inadequate and might mislead to a misunderstanding that all regulatory RNAs are involved in translational process. In addition, the manuscript does not contain any information regarding translation factor. The authors should exclude regulatory RNAs in the given sentence and add information regarding translation factor in translatomics.

Response: We thank the reviewer for this comment. To avoid misunderstanding, we explicitly stated that “not all regulatory RNAs are involved in translation” in section 2. Also, we added half a page for the translation factors in section 3.3.

3) In table 1, the authors should replace the description Easy/Hard with a more objective information. For example, “Operational difficulty” can be replaced with “Required experimental steps”.

Response: Thank you. Revised.

4) The authors should add legend for figure 1 and increase the size of its text.

Response: Thank you. We have rearranged the figures to increase the size of the text. We made a new figure specifically for nascent peptide analysis.

5) The authors should explain the figure 2 in more detail.

Response: Thank you. We added more details in the Fig.2 legend.

6) In “Application of translatomics”, the authors introduced translatomics application in various fields. It seems that this section could be separated into two. The authors should group 4.1, 4.2, 4.5, 4.6 as tranlsatomics in disease or stress condition for global translational detection and group 4.3, 4.4 as application of translatomics.

Response: Thank you. Motivated by the reviewer, we have partitioned the section 4 into two sections for biology/disease-relevant studies and applications, respectively.

7) The authors should describe each study in more detail in terms of the methods used to obtain results mentioned in “Application of translatomics”.

Response: Thank you. We have added the methods to this section.

7) In “Conclusion and Perspectives”, the authors should add more discussion regarding future perpective of translatomics and further improvements of the methods discussed in the manuscript.

Response: Thank you. We have added these discussions in to the Conclusion and Perspectives section.

8) The authors should improve English grammar thoroughly.

Response: Thank you. The English has been improved by a colleague who speaks very good English.

Reviewer 4 Report

This review from Zhao and Zhang overviews the main aspects of translatome research, with a focus on the techniques available and their pros and cons.

I very much appreciated the effort of putting together different views of the subject, from different standpoints, and I think that this kind of review is timely and useful for the researchers in the field. However, I think that a major limitation of the manuscript is related to the fact that the authors wanted to put too much in it, and some points are addressed too superficially. The result is a mixture of a little bit of this and a little bit of that, which in my opinion is not so useful. Instead, the authors should concentrate only on some of the touched matters, and look more deeply into them.

For instance, I think that the overview of the different techniques to study the translatome is extremely useful, but it should be enriched further. Just to give an idea of what I mean, no reference is made to the techniques exploiting FRET to study translation (just to cite one of these: FRET-Based Identification of mRNAs Undergoing Translation Benjamin Stevens , Chunlai Chen , Ian Farrell, Haibo Zhang, Jaskiran Kaur, Steven L. Broitman, Zeev Smilansky, Barry S. Cooperman, Yale E. Goldman  Published: May 31, 2012https://doi.org/10.1371/journal.pone.0038344).

In addition, as a general observation of this part of the manuscript, the authors should consider adding a brief hint about the possible applications of the described techniques, to create a sort of manual for those readers who are not so familiar with them (e.g., what answers can the techniques to study tRNAome answer?).

Also, the impact of ribosome diversity on translation regulation and on the translatome should be addressed, especially in inherited and acquired diseases.

Finally, I think that some of the parts included in “application of translatomics” should be removed, as the information given in them is not complete enough (i.e., the plants and circardian rhythm, and also the prokaryote part).

On top of these main considerations, I have some minor suggestions:

- some of the information included in table 1 does not match what is reported in the main text. For instance, polysome profiling is coupled to northern blotting, RT-PCR and microarray in the text, and to RNA-seq in the Table (which is correct). In addition, the last row of this table could be misleading. Indeed, the starting amount expressed in cell number depends on the cell type. With some cells, a lower number could be enough, whereas with others 5-10 folds more cells could be needed. I think that an indication of an amount of proteins in the loaded lysate could be more informative.

- The authors should also check the caption of figure 1, which is clearly wrong.

- There are several typos and probably the language should be checked by a mother tongue English speaker.

 I hope that the authors will find my thoughts and suggestions useful to improve their manuscript.

Author Response

Reviewer 4:

This review from Zhao and Zhang overviews the main aspects of translatome research, with a focus on the techniques available and their pros and cons.

 I very much appreciated the effort of putting together different views of the subject, from different standpoints, and I think that this kind of review is timely and useful for the researchers in the field. However, I think that a major limitation of the manuscript is related to the fact that the authors wanted to put too much in it, and some points are addressed too superficially. The result is a mixture of a little bit of this and a little bit of that, which in my opinion is not so useful. Instead, the authors should concentrate only on some of the touched matters, and look more deeply into them.

 For instance, I think that the overview of the different techniques to study the translatome is extremely useful, but it should be enriched further. Just to give an idea of what I mean, no reference is made to the techniques exploiting FRET to study translation (just to cite one of these: FRET-Based Identification of mRNAs Undergoing Translation Benjamin Stevens , Chunlai Chen , Ian Farrell, Haibo Zhang, Jaskiran Kaur, Steven L. Broitman, Zeev Smilansky, Barry S. Cooperman, Yale E. Goldman  Published: May 31, 2012https://doi.org/10.1371/journal.pone.0038344).

Response: We thank the reviewer for this information. In the revised manuscript, we added a section (2.6 Visualization of translation in vivo) to include this literature. We also included Nascent chain tracking (NCT) technique to monitor the translation. However, these methods are not high-throughput compatible.

In addition, as a general observation of this part of the manuscript, the authors should consider adding a brief hint about the possible applications of the described techniques, to create a sort of manual for those readers who are not so familiar with them (e.g., what answers can the techniques to study tRNAome answer?).

Response: We wrote many applications of the described techniques. Please refer to section 4 and 5. The question solved by tRNAome measurement is in the section 4.2.

 Also, the impact of ribosome diversity on translation regulation and on the translatome should be addressed, especially in inherited and acquired diseases.

Response: Thank you. We have added a section 3.5 for ribosome diversity.

 Finally, I think that some of the parts included in “application of translatomics” should be removed, as the information given in them is not complete enough (i.e., the plants and circardian rhythm, and also the prokaryote part).

Response: We thank the reviewer’s suggestions. Indeed, with increasing literatures published recently, it is hard to summarize all applications of translatomics within limited space. Since the reviewer 3 wanted us to arrange the application section, we have to leave this part in the manuscript. Also, this part provided examples of the problems that can be solved by various kinds of translatomics methods (as you mentioned above).

On top of these main considerations, I have some minor suggestions:

 - some of the information included in table 1 does not match what is reported in the main text. For instance, polysome profiling is coupled to northern blotting, RT-PCR and microarray in the text, and to RNA-seq in the Table (which is correct).

Response: Thank you. We have specified “high-throughput methods” in the revised Table 1.

In addition, the last row of this table could be misleading. Indeed, the starting amount expressed in cell number depends on the cell type. With some cells, a lower number could be enough, whereas with others 5-10 folds more cells could be needed. I think that an indication of an amount of proteins in the loaded lysate could be more informative.

Response: Thank you. We totally agree with the reviewers that different cell number is required for different cell types. To be on the safe side, we removed the last row of the Table 1.

 - The authors should also check the caption of figure 1, which is clearly wrong.

Response: Thank you. Revised.

 - There are several typos and probably the language should be checked by a mother tongue English speaker.

I hope that the authors will find my thoughts and suggestions useful to improve their manuscript.

Response: We deeply thank the reviewers for the thoughtful and insightful suggestions and made revisions. The language has been improved by a colleague.

 Round  2

Reviewer 1 Report

The manuscript has been improved according to my comments.

Response: Thank you.

Reviewer 3 Report

A majority of my original concerns were addressed by authors. The manuscript was significantly improved with additional and detailed informations.

Response: Thank you.

Reviewer 4 Report

I think that the present version of this review is very much improved compared to the original one, and offers to the reader a more comprehensive view of the available methods in translatomics. I just noticed a few typos in the new parts of the manuscript, please correct them.

Response: Thank you. Revised.